# Comprehensive In Silico Characterization and Expression Pro-Filing of DA1/DAR Family Genes in *Brassica rapa*

**DOI:** 10.3390/genes13091577

**Published:** 2022-09-02

**Authors:** Umer Karamat, Rui Yang, Yuhong Ren, Yin Lu, Na Li, Jianjun Zhao

**Affiliations:** State Key Laboratory of North China Crop Improvement and Regulation, Key Laboratory of Vegetable Germplasm Innovation and Utilization of Hebei, Collaborative Innovation Center of Vegetable Industry in Hebei, College of Horticulture, Hebei Agricultural University, Baoding 071000, China

**Keywords:** *Brassica rapa*, genome-wide, *DA1*&*DAR* family genes, expression profiling

## Abstract

The *DA1/DAR* family genes have been shown to play important roles in regulating organ size and plant biomass in the model plant Arabidopsis and several crops. However, this family has not been characterized in *Brassica rapa* (*B. rapa*). In this study, we identified 17 *DA1*&*DAR* genes from *B. rapa*. Phylogenetic analysis indicated that these genes are classified into four groups. Structural and motif analysis of *BrDA1*&*DARs* discovered that the genes within the same group have similar exon-intron structures and share an equal number of conserved motifs except for *BrDAR6.3* from group IV, which contains two conserved motifs. *Cis*-regulatory elements identified four phytohormones (Salicylic acid, Abscisic acid, Gibberellin, and auxin) and three major abiotic (Light, Low temperature, and drought) responsive elements. Further, six br-miRNAs named br-miR164a, br-miR164b, br-miR164c, br-miR164d, br-miRN360, and br-miRN366 were found which target *BrDAR6.1*, *BrDA1.4*, and *BrDA1.5*. *BrDA1*&*DAR* genes were highly expressed in stem, root, silique, flower, leaf, and callus tissues. Moreover, qRT-PCR analyses indicated that some of these genes were responsive to abiotic stresses or phytohormone treatments. Our findings provide a foundation for further genetic and physiological studies of *BrDA1*&*DARs* in *B. rapa*.

## 1. Introduction

The *DA1/DAR* gene family plays important roles in regulating organ size and plant biomass. DA1 was first identified in the model plant *Arabidopsis thaliana* (*A. thaliana*) as a key regulator for organ size [1]. In the *da1-1* mutant, the conversion of an amino acid from arginine to lysine at position 358 (*DA1^R358K^*) resulted in large leaves, flowers, siliques, and seeds due to an extended period of cell proliferation [1]. DA1 protein consists of two ubiquitin-interacting motifs (UIM), a zinc-binding LIM domain, and a C-terminal peptidase domain [1,2,3]. The Arabidopsis genome encodes seven DA1-related (DAR) proteins, among which DAR1 and DAR2 are the most closely related to DA1 based on amino acid sequences [1,4]. DAR1 acts redundantly with DA1 to regulate cell proliferation [1], while DA1, DAR1, and DAR2 redundantly regulate endoreduplication [4]. Recent studies have shown that the peptidase domain of DA1 and DAR proteins is important for its function on organ size regulation, and the peptidase activity is regulated by specific E3 ubiquitin ligases, de-ubiquitination enzymes, and receptor-like kinase [3].

In crops, the *DA1/DAR* family genes also function as key organ size and crop yield regulators. For instance, overexpression of the mutant version of *ZmDA1* or *ZmDAR1* enhances the starch synthesis and improves kernel yield in maize [5]. In wheat (*Triticum aestivum* L.), the expression level of *TaDA1* was correlated with kernel weight and yield [6]. In addition, BnDA1 is also associated with seed weight in *Brassica napus* (*B. napus*) [7]; down-regulation of BnDA1 improves the seed weight and organ size [7]. These studies suggested that the function of *DA1* family genes in regulating organ size is highly conserved in different species and may be used as a molecular target for high-yield breeding.

Since the *DA1*&*DAR* gene family plays regulatory roles in developmental processes in plants, we wanted to identify *BrDA1*&*DAR* genes in *B. rapa* and investigate their involvement in different stress conditions. *B. rapa* is a diverse plant species belonging to Cruciferae (Brassicaceae) family, which is further divided into three well-defined groups named oil-type rape, leafy-type *B. rapa*, and rapiferous-type [8]. There have been no studies about the characterization and stress response behavior of the *DA1*&*DAR* gene family in *B. rapa*. We conducted a comprehensive genome-wide study to identify the *DA1*&*DAR* gene family in *B. rapa*. We identified 17 *DA1*&*DAR* genes in the *B. rapa* genome and analyzed the phylogenetic relationships, synteny examination, gene structures, conserved motifs, *cis*-elements in the promoter region, and miRNA regulator prediction. Additionally, we studied *DA1*&*DAR* gene expression in various tissues and under different abiotic and phytohormone treatments. These analyses are helpful for further functional characterization of the *DA1*&*DAR* gene family in *B. rapa*.

## 2. Materials and Methods

### 2.1. Identification and Characterization of DA1&DARs in B. rapa

Protein sequences of all *BrDA1*&*DARs* were obtained via accessing the *Brassica* database (http://www.brassicadb.cn/ (Version 3, 3.1 and 3.5), accessed on 10 June 2022). The amino acid sequences of *AtDA1*&*DARs* were downloaded from the *A. thaliana* database (http://www.arabidopsis.org/, accessed on 10 June 2022). Peptide sequences of *AtDA1*&*DARs* were used as query sequences to identify all *BrDA1*&*DARs*. Then (http://pfam.xfam.org/, accessed on 10 June 2022) was accessed to scrutinize the DA1-like domain (PF12315) in all sequences and verify that all recognized genes from the *Brassica* database belong to the *BrDA1*&*DAR* gene family. PROTPARAM on ExPASy (http://web.expasy.org/protparam/, accessed on 10 June 2022) was accessed to measure chemical and physical properties such as isoelectric points (PI) and molecular weight (MW). To predict the sub-cellular localization of *BrDA1*&*DAR* genes WoLF PSORT server (https://wolfpsort.hgc.jp/, accessed on 10 June 2022) was accessed. Gene Structure Display Server 2.0 (http://gsds.gao-lab.org/, accessed on 10 June 2022) was used to analyze the gene structure of *BrDA1*&*DAR* genes. To identify the conserved motifs, MEME (V 4.11.4) was used.

### 2.2. Analysis of Phylogenetic Trees and Syntenic Pairing BrDA1&DAR Family Proteins

To observe and analyze the evolutionary background, the *DA1*&*DAR* family proteins of *B. rapa*, *A. thaliana*, *B. napus*, and *B. oleracea* were used to construct the phylogenetic tree. Alignment of protein sequences was performed using MEGA X (V6.06) software. The tree was constructed via the neighbor-joining (NJ) method with 1000 bootstrap replicates. Synteny associations of *DA1*&*DAR* genes between *B. rapa*, *A. thaliana*, *B. oleracea*, and *B. napus* were identified and analyzed via TBtools software V1.098; (https://github.com/CJ-Chen/TBtools, accessed on 10 June 2022).

### 2.3. Identification and Analysis Cis-Elements in the BrDA1&DAR Gene Promoters

Next, 2000 base pairs upstream of *BrDA1*&*DAR* genes were downloaded from the Brassica database (http://www.brassicadb.cn/, accessed on 10 June 2022) to find the *cis*-regulatory element. PlantCARE webtool (http://bioinformatics.psb.ugent.be/webtools/plantcare/html/, accessed on 10 June 2022) was accessed to predict the *cis*-elements, then results were presented using TBtools software V 1.098; (https://github.com/CJ-Chen/TBtools, accessed on 10 June 2022).

### 2.4. Prophecy of Putative miRNA Targeting BrDA1&DAR Genes

*BrDA1*&*DAR* gene sequences were used as candidate genes to identify predicted miRNAs via the psRNATarget database (http://plantgrn.noble.org/psRNATarget/, accessed on 10 June 2022) with default parameters. The interaction network between the predicted miRNAs and the target *BrDA1*&*DAR* genes was created using Cytoscape V3.8.2 software (https://cytoscape.org/, accessed on 10 June 2022).

### 2.5. Expression Profiling of BrDA1&DAR Genes in Various Tissues

We used RNA-seq data [9] to highlight the expression patterns of *BrDA1*&*DAR* genes. Expression was analyzed from six tissues (root, stem, leaf, callus, flower, and silique) of *B. rapa* accession Chiifu-401-42. Expression values were analyzed in FPKM (fragments per kilobase of exon model per million mapped reads). We identified and generated the heatmap of expression of *BrDA1*&*DAR* genes using GraphPad Prism 9.0.0 software (https://www.graphpad.com/, accessed on 10 June 2022).

### 2.6. Plant Material and Stress Conditions

Wild-type Chinese cabbage “A03” was grown under different stress conditions. Seeds with 100% germination were selected and the growing conditions were 25 °C Day/Night and 16 h light and 8 h dark cycle. The effects of phytohormones on the seedlings were determined by treating them with 100 µM gibberellic acid (GA), 100 µM abscisic acid (ABA), 100 µM salicylic acid (SA), and 100 µM indole-acetic acid/auxin (IAA). The samples were collected at successive 2 h after the application of stress, i.e., 0 h (CK as control), 2 h, 4 h, 6 h, 8 h. Polyethylene Glycol PEG6000 15% solution was applied for the imposition of drought stress and samples were collected at 0 h (CK), 2 h, 4 h, 6 h, 8 h. For salt stress, NaCl 250 mM solution was applied, and samples were collected at 0 h (CK), 6 h, 12 h, 18 h, 24 h after the treatment. Samples were collected at 4 °C for cold stress analysis, and for heat stress, the temperature was set at 44 °C. After the stress imposition, samples were collected at 0 h (CK), 1 h, 3 h, 6 h, 12 h, and 24 h. the leaves were immediately immersed in liquid nitrogen and kept stored at −80 °C until further analysis.

### 2.7. RNA Extraction and qRT-PCR Analysis

For RNA extraction, Eastep^®^ Super RNA Isolation Kit, Promega Biotech, Shanghai, China was used according to the protocol provided by the manufacturer. Then the quantity of extracted RNA was measured on Nanodrop One (Thermo Fisher Scientific, Worcester, MA, USA). The cDNA Synthesis SuperMix kit (TransGen Biotech, Beijing, China) was used for cDNA synthesis. Afterward, deionized water was added to dilute the cDNA into a 10× solution. qRT-PCR was performed with a CFX Connect Real-Time System (Bio-Rad, Hercules, CA, USA), using SYBER^®^ Green Supermix (Bio-Rad). *BrActin* primers were used as a control to analyze the results. The qRT-PCR reaction was performed as follows: 94 °C for 10 min, followed by 40 cycles of 94 °C for 15 s, 60 °C for the 30 s, and 72 °C for 10 s. Each reaction was performed with three biological replicates, and then all the results were analyzed by using the 2^−ΔΔCT^ method as described. All the primers used in this experiment are listed in Appendix A.

## 3. Results

### 3.1. Identification of BrDA1&DAR Gene Family in B. rapa

Using AtDA1&DAR protein sequences as a query, BLASTP identified 17 *DA1*&*DAR* genes in the complete genome of *B. rapa* (Table 1). These genes were named *BrDA1.1*-*BrDA1.5*, *BrDAR1.1*-*BrDAR1.3*, *BrDAR2.1*-*BrDAR2.5*, *BrDAR3.1* and *BrDAR6.1*-*BrDAR6.3*. We submitted amino acid sequences of all the BrDA1&DARs to the NCBI (https://www.ncbi.nlm.nih.gov/Structure/cdd/cdd.shtml, accessed on 10 June 2022). We then analyzed the results in TBtools software V 1.098 to confirm the presence of the DA1-like domain in all identified genes (Appendix A). Furthermore, we accessed http://pfam.xfam.org/ to verify the DA1-like domain (PF12315) in all sequences. Gene length varies from 2208 bp (*BrDA1.3*) to 8063 bp (*BrDAR1.2*) with 9 and 13 exons, respectively, and the highest number of exons was found in *BrDAR1.1*, which is 22. The coding sequences (CDS) varied from 462 bp (*BrDAR6.3*) to 2929 bp (*BrDAR1.2*), while the protein size varied from 154 (BrDAR6.3) to 976 (BrDAR1.2). The molecular weight of all BrDA1 proteins ranged from 17.11 kDa (*BrDAR6.3*) to 110.31 (*BrDAR1.2*). The subcellular localization predicted that 13 BrDA1&DAR proteins are localized on the nucleus, two on the cytoplasm, one on the endoplasmic reticulum, and one on the peroxisome. Meanwhile, 13 *DA1*&*DAR* family genes from *Brassica oleracea* (*B. oleracea*) and *B. napus* were also identified (Appendix A). 

### 3.2. Phylogenetic Relationships of BrDA1&DAR Genes

To understand the evolution of *BrDA1*&*DARs*, *AtDA1*&*DARs*, *BnDA1*&*DARs*, and *BolDA1*&*DARs*, an unrooted phylogenetic tree was established that was further divided into four groups (Groups I–IV) (Figure 1). Detailed results showed Group I contained 12 *DAR2* family members (5 *BrDAR2*, 4 *BnDAR2*, 2 *BolDAR2*, and 1 *AtDAR2*); Group II consisted of 6 *DAR1* (3 *BrDAR1*, 1 *BnDAR1*, 1 *BolDAR1*, and 1 *AtDAR1*); Group III had 12 *DA1* members (5 *BrDA1*, 4 *BnDA1*, 2 *BolDA1*, and 1 *AtDA1*) and Group IV comprised of 19 members including 4 *DAR3*, 1 *DAR4*, 3 *DAR5*, 10 *DAR6*, and 1 *DAR7*gene. Notably, Groups I and III had more *BrDA1*&*DAR* genes than the other two groups. Furthermore, it was found that *BrDA1*&*DAR* genes are closer to *BnDA1*&*DARs* and *BolDA1*&*DARs*.

### 3.3. Synteny Analysis of BrDA1&DAR Genes

Tandem and segmental duplication always support plant genome progression and the development of gene family members. The mechanism segmental and tandem duplication developments were examined in *BrDA1*&*DAR* genes. The distribution of 16 *BrDA1*&*DAR* genes on chromosomes was evaluated in which 6 out of 10 chromosomes contained *BrDA1*&*DAR* genes (Figure 2). Chromosomes A01, A05, A06, and A09 possess three *BrDA1*&*DARs*, while A03 and A08 have two genes. The remaining chromosome did not contain any *DA1*&*DAR* genes. However, no paralogous gene was found on any chromosome. All *BrDA1*&*DAR* genes were found as evolved through tandem duplication. No proximal duplication was found on any chromosomes (Figure 2). These findings showed that tandem duplication actions played an important role in the expansion of the *DA1*&*DAR* gene family in *B. rapa*.

Collinearity analysis was performed to discover orthologs of *DA1*&*DARs* between *B. rapa*, *A. thaliana*, *B. oleracea*, and *B. napus* (Figure 3). Briefly, In *B. rapa*, the sequences of 17 genes belonging to *BrDA1*&*DARs* were collected from the latest genome assemblies V3, 3.1, 3.5. However, there were only 6 *BrDA1*&*DARs* that showed syntenic relationships with 6 *AtDA1*&*DARs*, 7 *BolDA1*&*DARs*, and 12 *BnDA1*&*DARs* present in V3.0 (http://www.brassicadb.cn/, accessed on 10 June 2022). In detail, one *AtDA1* from Chr1 is associated with *BrDA1.1* gene of A06 and *BrDA1.4* on A08, one *AtDAR2* from Chr2 is related to *BrDAR2.1* gene of A05, *AtDAR1* from Chr4 is associated to one *BrDAR1.2* gene on A01, and *AtDAR6* gene from Chr5 made syntenic association with one *BrDAR6.1* gene on A07 and *BrDAR3.1* on A09. The syntenic relationship between the *DA1*&*DAR* family of *B. rapa*, *B. oleracea*, and *B. napus* is presented in Figure 3.

Ka, Ks, and Ka/Ks ratio was measured to study the evolutionary process of *BrDA1*&*DAR* genes, and less than 1 Ka/Ks ratio was found for all duplicated genes, suggesting that the *DA1*&*DAR* gene family has been facing the discriminatory burden and selection pressure throughout its evolutionary process (Appendix A).

### 3.4. Gene Structure and Conserved Motifs Analysis of BrDA1&DAR Gene Family

To enhance our understanding of gene development of the *B. rapa DA1*&*DAR* family, exon-intron configuration was observed. Introns of the *BrDA1*&*DARs* ranged from 2 to 21 (Table 1; Figure 4A). Group I has 7 to 11 introns while Group II has the highest 11 to 21, Group III has 7 to 8 introns, and Group IV has 2 to 12 introns. *BrDAR1.1* from group II has the highest introns, while *BrDAR6.3* from group IV has the lowest introns, only 2. Group I has 8 to 12 exons, Group II has the highest number of exons ranging from 12 to 22, Group III has 8 to 9 exons, and Group IV has 3 to 13 exons. Intron-exon patterns were similar between Groups I and III, whereas Groups II and IV exhibited different intron/exon associations. These results indicate that most of the same group members had identical gene structures, supporting their phylogenetic relations.

We also studied protein sequences of all 17 BrDA1&DARs through MEME to identify the preserved region and found that their conserved motif of BrDA1&DAR proteins varies from 2 to 8. Motif distributions within groups were similar, especially in Groups I, II and III (Figure 4B). Group II and III members have eight conserved motifs except for BrDAR1.3 from Group II, which has seven conserved regions. Conversed motifs in Group I ranged from 5 to 8, a single member from Group I has five conserved regions, and Group IV possesses 5 to 6 conserved motifs except for BrDAR6.3, which has only two conserved regions only (Appendix A). The group classification based on phylogenetic relationship, gene structure, and conserved region analysis strongly supported that BrDA1&DAR proteins have identical peptide remains and that most of the members of the same group have similar roles.

### 3.5. Cis-Elements in Promoters of BrDA1&DAR Genes

We searched 2000 bp regions from the transcriptional active site of each gene against the PlantCARE database to identify *cis*-elements in their promoter region (Appendix A). Among four different hormones responsive elements SA, ABA, and IAA-related elements were widely distributed among different *BrDA1*&*DAR* genes showing their unique and vital role in phytohormone-regulated plant development and stress responses (Figure 5; Appendix A). Besides this, four different stress-responsive regulatory elements, such as anaerobic, drought, light, and low temperature, were also identified, indicating the stress-responsiveness of these genes (Figure 5; Appendix A). Identification of hormones and stress-related *cis*-elements indicates that expression profiling of *BrDA1*&*DAR* genes may vary under hormonal and abiotic stress conditions.

### 3.6. Identification of miRNA Targeting Sites in BrDA1&DAR Genes

During the last few years of innovative research, many investigations have found that miRNA mediates regulations of different genes under some stress conditions. We found six putative miRNAs that target three *BrDA1*&*DAR* genes (Figure 6). The complete information is available (Figure 6, Appendix A). Among them, four belong to the br-miR164 family that targets the *BrDAR6.1* gene, whereas br-miRN366 and br-miRN360 target *BrDA1.4* and *BrDA1.5* in *B. rapa* (Figure 6).

### 3.7. Expression Profiling of BrDA1&DAR Gene Family

To demonstrate the expression of *BrDA1*&*DARs*, we used the RNA-Seq data of six different tissues that were analyzed by Tong [9]. The six genes (*BrDA1.1*, *BrDA1.4*, *BrDAR1.2*, *BrDAR2.1*, *BrDAR2.2*, and *BrDAR3.1*) are from Version 3, which has undergone transcriptome analysis. Other genes are from the upgraded versions 3.1 and 3.5, for which no transcriptome or expression data is currently available. The transcript profiling of these six *BrDA1*&*DAR* genes from root, stem, leaf, callus, flower, and silique showed that *BrDA1.1*, *BrDA1.4*, and *BrDAR1.2* were highly expressed in all tissues, especially in root, flower, silique, and callus, which reflected their significant role in growth processes of *B. rapa* (Figure 7).

### 3.8. Expression Analysis of BrDA1&DAR Genes under Hormone Treatment and Abiotic Stresses

Upon hormone treatment, most of the genes showed relatively high expression in response to ABA, while most genes showed relatively low expression under SA, IAA, and GA (Figure 8). For instance, in response to ABA, *BrDA1.1*, and *BrDA1.4* at 6 h, *BrDAR1.1* and *BrDA2.3* at 4 h, *BrDA2.1* at 8 h, *BrDA2.2*, *BrDA2.5*, *BrDA6.1* at 4 h and 6 h and *BrDA3.1* at 6 h and 8 h were upregulated. Under salicylic acid (SA), *BrDA1.1* at 4 h and 8 h, *BrDA1.3* at 8 h, *BrDAR2.4* and *BrDAR2.5* at 2 h, *BrDAR6.2* and *BrDAR6.3* at 2 and 8 h were found to be highly expressed. While *BrDAR1.1*, *BrDAR1.2*, *BrDAR2.2*, *BrDAR6.1*, and *BrDAR6.2* showed high expression under IAA at different time intervals. In response to SA, only four genes (*BrDA1.2*, *BrDA1.3*, *BrDAR6.2*, and *BrDAR6.3*) were upregulated at different time points, as shown in Figure 8. However, the remaining *BrDA1*&*DARs* did not show any significant difference under phytohormone stress (Figure 8).

Interestingly, most genes showed varied expression levels under cold and high temperatures (Figure 9). *BrDA1.3*, *BrDA1.5*, and *BrDAR3.1* were most upregulated under low temperature (4 °C) compared to others, while *BrDA1.4*, *BrDAR2.1*, *BrDAR6.1*, and *BrDAR6.3* were highly expressed at high-temperature treatment (44 °C).

Relatively few *BrDA1*&*DAR* genes showed high expression under drought and salinity stress as compared to phytohormones and temperature stresses. In detail, *BrDA1.3*, *BrDA1.5*, *BrDAR1.1*, *BrDAR1.2*, *BrDAR2.2*, *BrDAR2.3*, *BrDAR6.2*, and *BrDAR6.3* were upregulated at different time points under drought stress while remaining genes showed minimum expression (Figure 10a). Under salinity stress, *BrDA1.2*, *BrDA1.3*, *BrDA1.5*, *BrDAR1.1*, *BrDAR2.2*, *BrDAR2.4*, *BrDAR6.2*, and *BrDAR6.3* were upregulated as compared to other genes (Figure 10b).

## 4. Discussion

In this study, we identified 17 BrDA1&DAR proteins containing DA1-like domain [10,11]. Moreover, we identified 13 from *B. oleracea* and 11 from *B. napus* (Table 1; Appendix A). Whole genome sequencing plays a crucial role in developing a strong understanding of evolution, domestication, and diversification, as demonstrated by whole-genome triplication (WGT) of *Brassica* species, such as *B. rapa* and *B. oleracea* [12]. Among *Brassica* plants, many species display extreme traits, and each one underwent a WGT event when compared with *A. thaliana*. Although many of the paralogous genes derived from the WGT event were fractionated (loss of a duplicate), the conserved ones probably play a significant role in the domestication and the diversification of phenotypic characters [13]. According to phylogenetic analysis, all *DA1*&*DARs* were classified into four different groups (Figure 1). Group I and III have more *BrDA1*&*DARs*, as both shared 5 genes, while Group II possesses three and Group IV contains 4 *BrDA1*&*DARs* (Figure 1). The presence of all *DA1*&*DARs* in four groups shows their strong phylogenetic relationship, implying their close evolutionary correlation (Figure 1). Further investigation of gene structure analysis also supports these results, as *BrDA1*&*DAR* genes from the same groups have a conserved number of introns (Figure 4). Furthermore, motif regions within the group classification were comparable (Figure 4B). Notably, *BrDAR6.3* from group IV has a diverse motif pattern compared to other members, suggesting its unique role in physiological and molecular functions as a growth regulator (Figure 4B). Proteins with strong evolutionary relationships and sharing similar motif composition suggest quite a similar function of *BrDA1*&*DAR* genes within the group.

Expression analysis of six *BrDA1*&*DAR* genes in various tissues such as root, stem, leaf, callus, flower, and silique suggested that most of the *BrDA1*&*DAR* genes are highly expressed in root, flower, silique, and callus while *BrDA1.1*, *BrDA1.4*, and *BrDAR1.2* were found to be expressed strongly in all tissues In *A. thaliana*, *AtDA1*, *AtDAR1*, *AtDAR2*, and *AtDAR4* were expressed highly in all tissues whereas *AtDAR3* showed high expression in root and leaf, *AtDAR5* and *AtDAR6* highly expressed in root and stem and *AtDAR7* showed maximum expression in flowers tissue (http://www.brassicadb.cn/#/Transcriptome/). *B. rapa AtDA1* expression is affected by ABA stress, while *AtDAR4* is associated with cold stress [1,14]. In soybean (*Glycine max*), the *GmaDA1* gene responded to drought, salt, ABA, alkali, and acid stress [15]. In this study, we found that *BrDA1.1*, *BrDA1.4*, *BrDAR1.1*, *BrDAR2.3*, *BrDAR2.1*, *BrDAR2.2*, *BrDAR2.5*, *BrDAR6.1*, *BrDAR3.1* were mainly upregulated by ABA stress as compared to other hormones (Figure 8) and all these genes have abscisic acid-responsive elements in their promoter region (Figure 5). Under GA stress, *BrDAR2.2*, *BrDAR6.1*, and *BrDAR6.3* were upregulated (Figure 8), while only *BrDAR2.2* and *BrDAR6.3* have GA-responsive elements in their promoter region (Figure 5). In response to SA, only *BrDA1.2*, *BrDA1.3*, *BrDAR2.4*, *BrDAR6.2*, and *BrDAR6.3*, expressed highly (Figure 8), and all genes have SA-related regulatory elements except *BrDAR6.2*, and *BrDAR6.3* (Figure 5). While under IAA treatment, *BrDA1.5*, *BrDAR1.2*, *BrDAR2.2*, and *BrDAR6.1* were upregulated (Figure 8), and only *BrDAR1.2* and *BrDAR6.1* contain IAA responsive elements in their promoter region (Figure 5). Interestingly, almost all members of the *BrDA1*&*DAR* gene family were upregulated by cold and heat stress (Figure 9). Still, in our *cis*-responsive element analysis, only *BrDAR2.2*, *BrDAR6.2*, and *BrDAR6.3* have low temperature-related *cis*-element in their promoter region (Figure 5). While few members of the *BrDA1*&*DAR* gene family such as *BrDA1.3*, *BrDA1.5*, *BrDAR1.1*, *BrDAR1.2*, *BrDAR2.2*, *BrDAR2.3*, *BrDAR6.2*, and *BrDAR6.3* were highly expressed under drought tolerance, out of these genes, promoter regions of three genes named *BrDA1.5*, *BrDAR6.2*, and *BrDAR6.3* contain drought-responsive elements (Figure 5), and *BrDA1.2*, *BrDA1.3*, *BrDA1.5*, *BrDAR1.1*, *BrDAR2.2*, *BrDAR2.4*, *BrDAR6.2*, and *BrDAR6.3* were found to be upregulated by salinity stress (Figure 10a,b). These results indicate that the *BrDA1*&*DAR* gene family is involved in stress response.

Recently, many miRNAs have been identified and investigated in *B. rapa* that were involved in gene regulatory networks and different environmental stresses [16,17,18]. Present studies identified the four members of the br-miR164 family targeting one *BrDAR6.1* gene, while br-miRN366 and br-miRN360 were found to target and regulate the two *DA1* genes in *B. rapa* named *BrDA1.4* and *BrDA1.5* (Figure 6). Moreover, br-miR164 family members were involved in regulating heat tolerance in Chinese cabbage [19,20,21]. Our results indicated that the expression of *BrDAR6.1* got upregulated under heat stress, and the maximum expression was at 12 h. It would be interesting to investigate whether the br-miR164-*BrDAR6.1* module functions in heat response.

## Figures and Tables

**Figure 1 genes-13-01577-f001:**
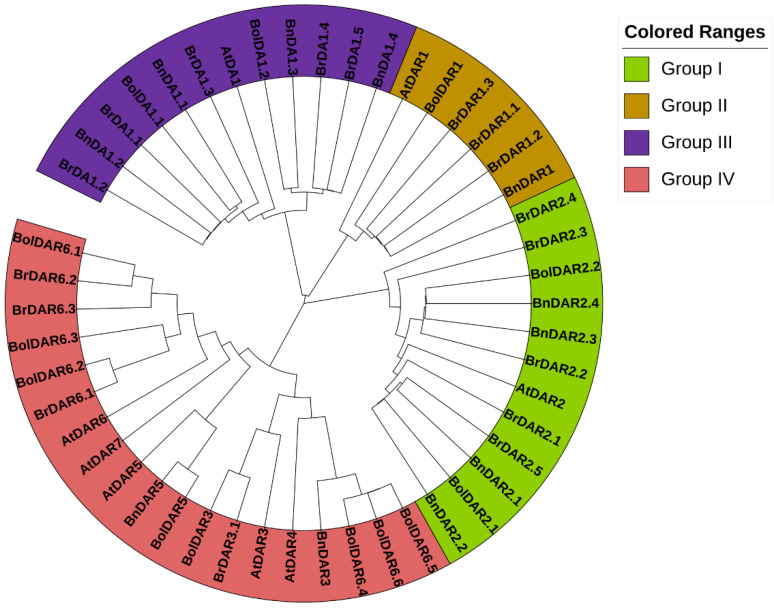
Phylogenetic analysis of 17 *DA1*&*DAR* proteins from *B. rapa*, 11 from *B. napus*, 13 from *B. oleracea*, and 7 from *A. thaliana*.

**Figure 2 genes-13-01577-f002:**
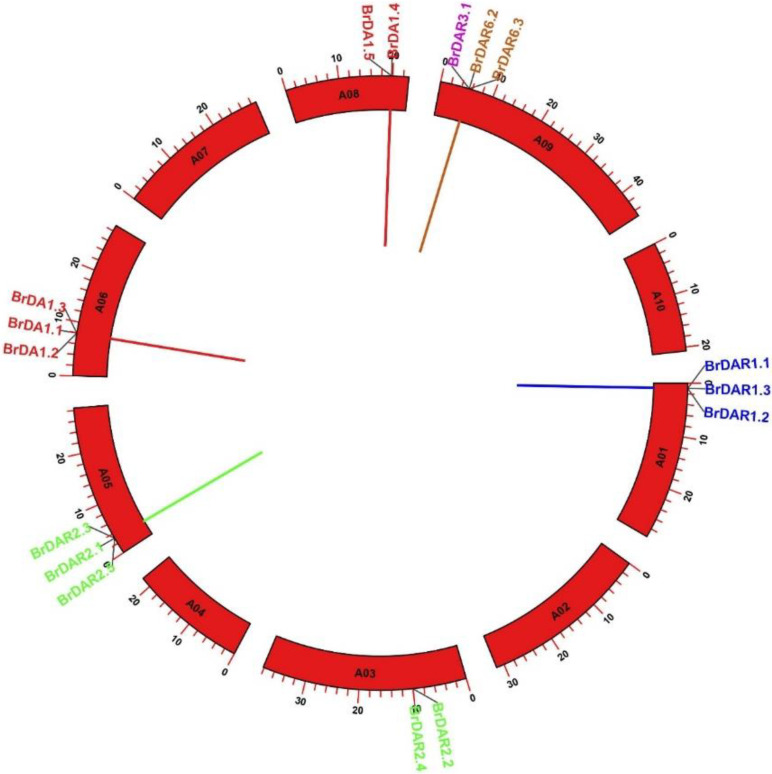
Circular representation of chromosomal distribution and interaction of *BrDA1*&*DARs* evolved through tandem duplication. Based on phylogenetic analysis, different colors represent the presence of *BrDA1*&*DAR* genes in different groups.

**Figure 3 genes-13-01577-f003:**
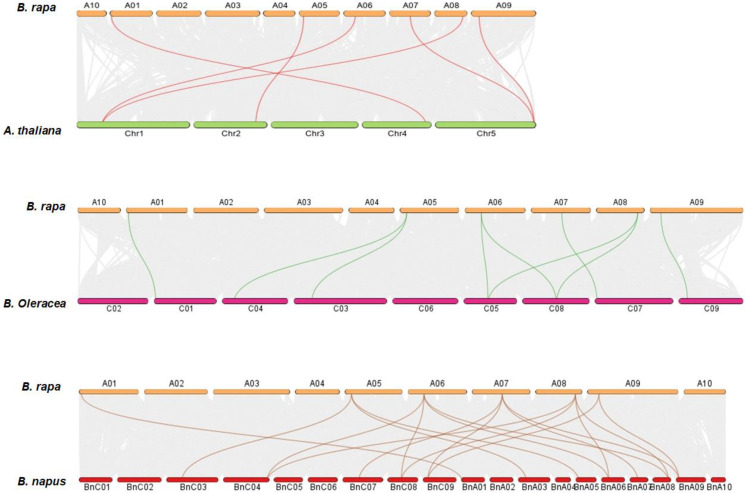
Synteny analysis of *DA1*&*DARs* between *B. rapa*, *A. thaliana*, *B. oleracea*, and *B. napus*. Gray lines show the syntenic pairing between *B. rapa*, *A. thaliana*, *B. oleracea*, and *B. napus* genomes, while colored lines represent the syntenic association.

**Figure 4 genes-13-01577-f004:**
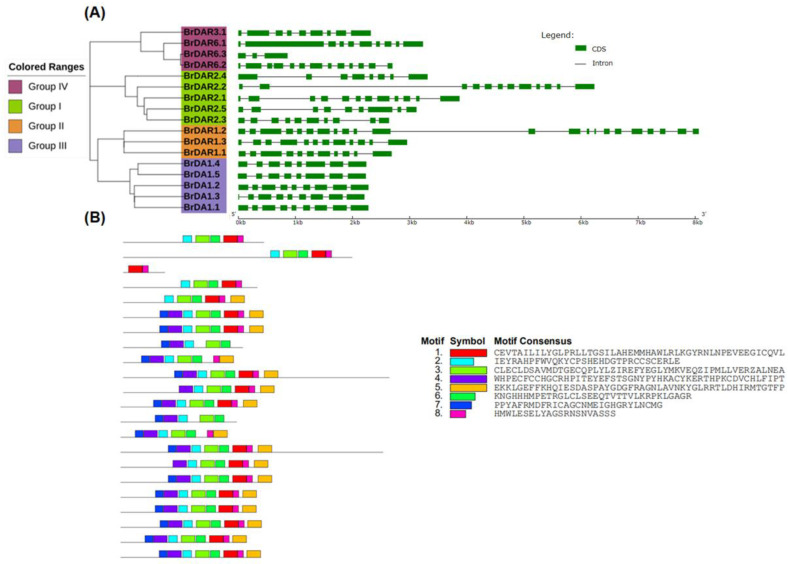
Structural and motif analysis of *BrDA1*&*DAR* genes family. According to the phylogenetic analysis, the *BrDA1*&*DAR* genes were classified into four groups. (**A**) Gene structure analysis. (**B**) The conserved motifs of BrDA1&DARs.

**Figure 5 genes-13-01577-f005:**
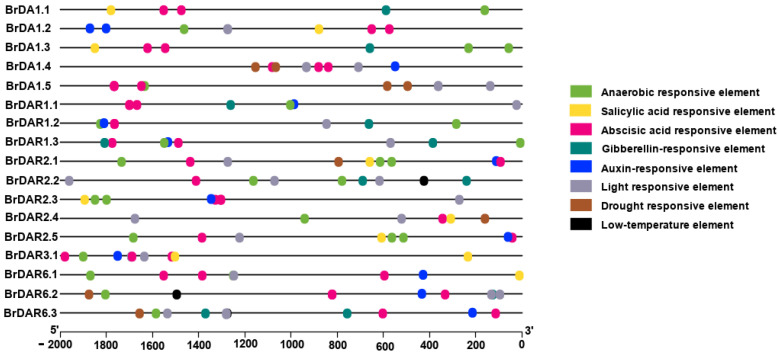
Different abiotic stress (anaerobic, light, drought, low temperature) and phytohormone (SA, ABA, GA, IAA) related *cis*-regulatory elements in *BrDA1*&*DAR*. Boxes of different colors show the different identified elements.

**Figure 6 genes-13-01577-f006:**
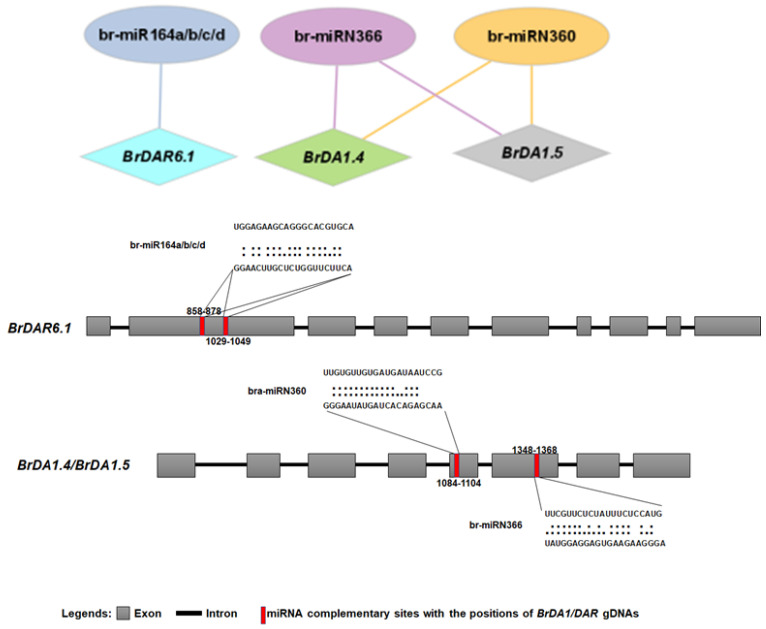
The miRNA targeting sites in *BrDA1*&*DAR* genes (*BrDAR6.1*, *BrDA1.5*, and *BrDA1.4*).

**Figure 7 genes-13-01577-f007:**
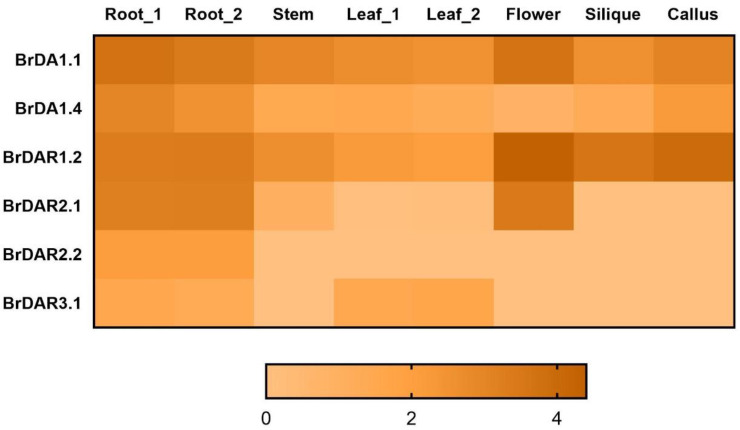
Heatmap presentation of *BrDA1*&*DARs* genes expression in various tissues (root, stem, leaf, callus, flower, and silique). The bar at the bottom presents the expression value.

**Figure 8 genes-13-01577-f008:**
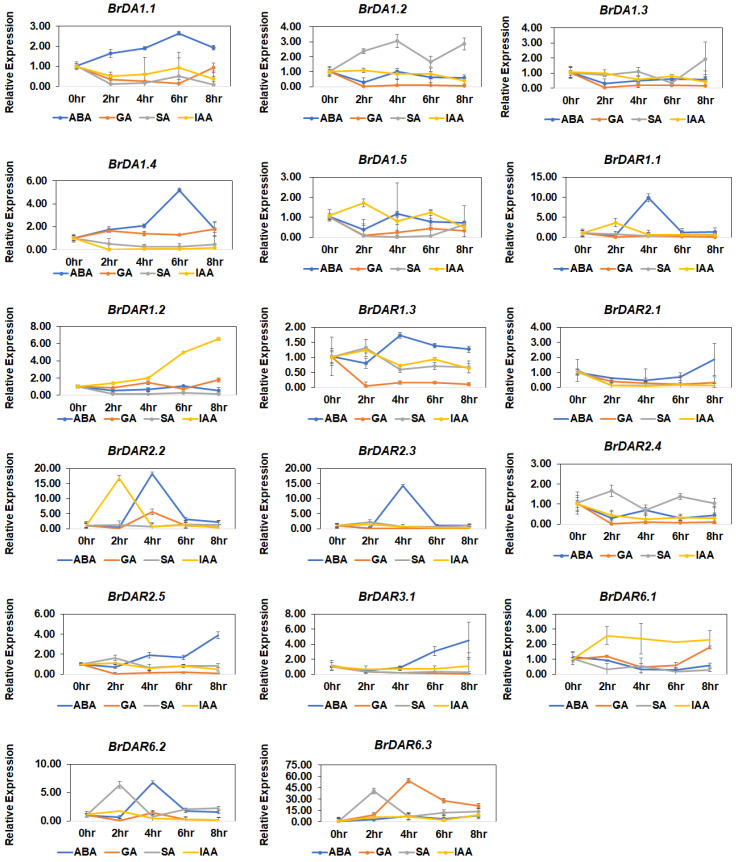
Expression profiling of *BrDA1*&*DAR* under different phytohormones (ABA, GA, SA, and IAA). 0 h, 2 h, 4 h, 6 h, and 8 h are the time intervals of sampling after treatment, and graphs represent relative gene expression examined through the 2^−ΔΔCT^ method.

**Figure 9 genes-13-01577-f009:**
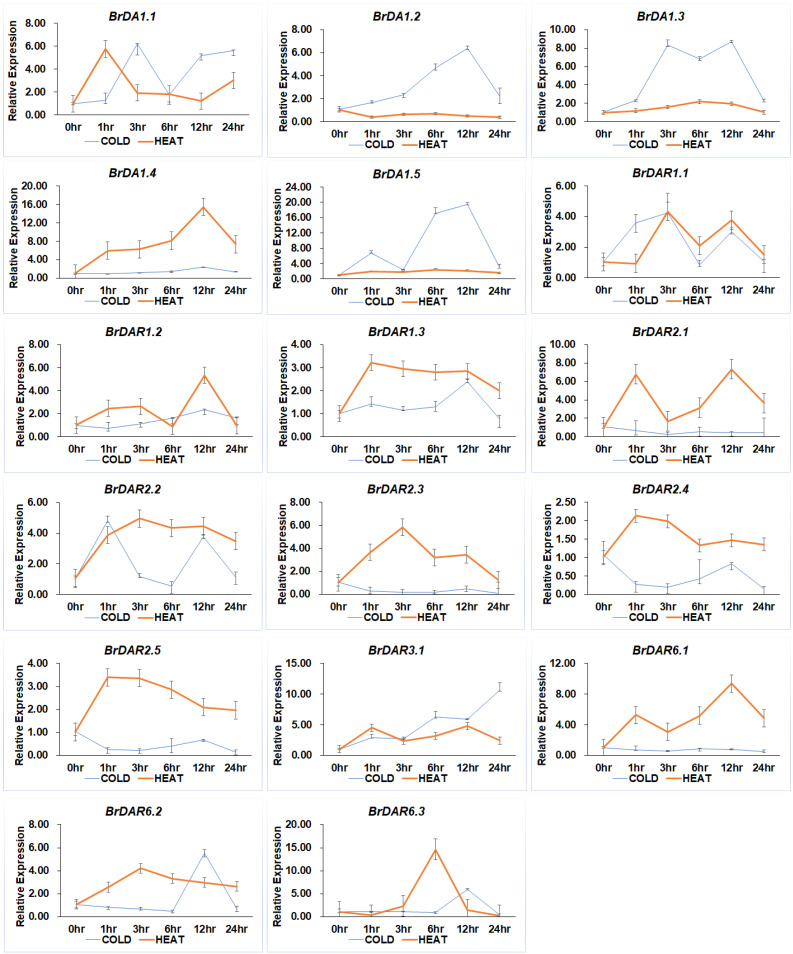
Relative expression patterns of *BrDA1*&*DAR* genes under cold (4 °C) and high temperature (44 °C) treatment. Time points of 0 h as CK, 1 h, 3 h, 6 h, 12 h, and 24 h displayed the sampling intervals.

**Figure 10 genes-13-01577-f010:**
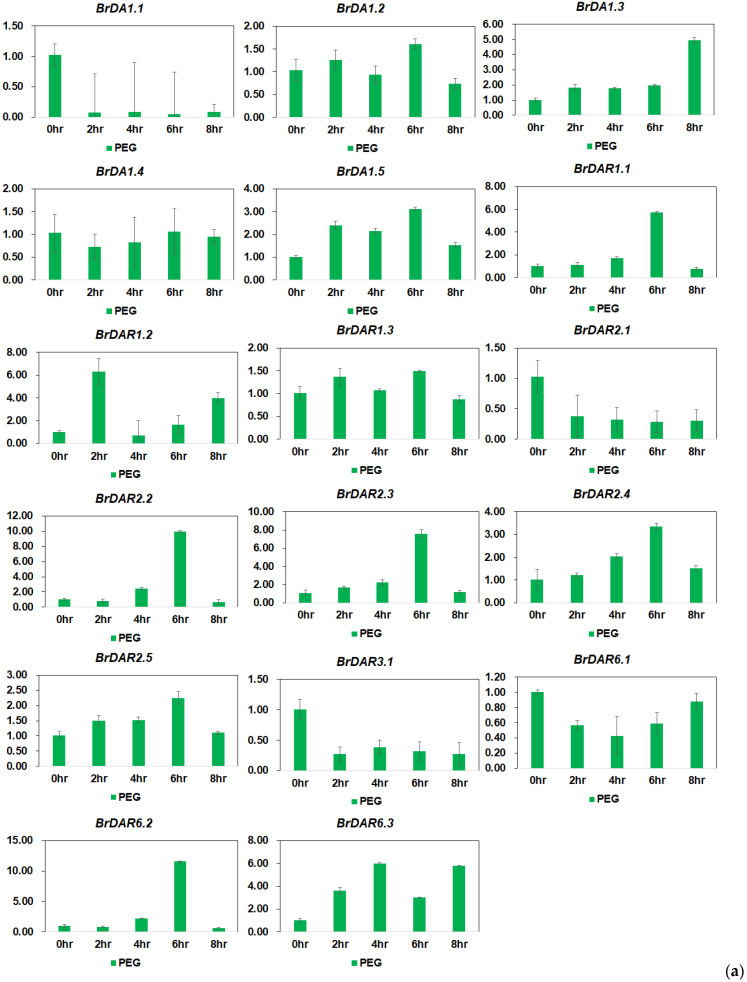
(**a**) Expression profiling of *BrDA1*&*DAR* genes under drought stress. Sampling was performed at 0 h (CK), 2 h, 4 h, 6 h, and 8 h. 2^−ΔΔCT^ method was used to examine the results. (**b**) Expression patterns of *BrDA1*&*DAR* genes under salinity stress. Sampling was performed at 0 h (CK), 6 h, 12 h, and 24 h. 2^−ΔΔCT^ method was used to analyze the results.

**Table 1 genes-13-01577-t001:** Detailed information of 17 *BrDA1*&*DAR* gene family.

Transcript ID	ID in AT	Name Found on Database	Given Name	Genomic Location	Gene/CDS Length (bp)	Protein Length (AA)	Protein Molecular Weight (kDa)	Isoelectric Point (pI)	No of Exon/intron	Predicted Sub-Cellular Localization
BraA06g014880.3C	AT1G19270	DA1	BrDA1.1	A06:7,920,264−7,922,541+	2278/1584	528	59.68	5.89	9/8	Nucleus
BraA06g015110.3.5C	AT1G19270	DA1	BrDA1.2	A06:7,919,363−7,922,691+	3329/1593	531	59.93	5.89	9/8	Nucleus
BraA06g015150.3.1C	AT1G19270	DA1	BrDA1.3	A06:7,920,334−7,922,541+	2208/1428	476	53.67	6.38	9/8	Nucleus
BraA08g028280.3C	AT1G19270	DA1	BrDA1.4	A08:19,778,813−19,781,054−	2242/1539	513	58.56	6.06	8/7	Nucleus
BraA08g028910.3.5C	AT1G19270	DA1	BrDA1.5	A08:19,778,513−19,781,625−	3113/1536	512	58.6	6.06	8/7	Nucleus
BraA01g001960.3.5C	AT4G36860	DAR1	BrDAR1.1	A01:9,92,491−9,96,203+	3713/1704	568	64.53	4.99	22/21	Nucleus
BraA01g001980.3C	AT4G36860	DAR1	BrDAR1.2	A01:9,93,314−1,001,376+	8063/2929	976	110.31	5.22	11/10	Nucleus
BraA01g001960.3.1C	AT4G36860	DAR1	BrDAR1.3	A01:9,93,037−9,95,992+	2956/1662	554	62.89	5.62	12/11	Cytoplasm
BraA05g006240.3C	AT2G39830	DAR2	BrDAR2.1	A05:3,156,572−3,160,445+	3874/1545	515	58.24	8.55	8/7	Nucleus
BraA03g021030.3.5C	AT2G39830	DAR2	BrDAR2.2	A03:10,048,165−10,054,720−	6556/1545	515	57.94	7.09	12/11	Nucleus
BraA05g006190.3.1C	AT2G39830	DAR2	BrDAR2.3	A05:3,157,807−3,160,445+	2639/1218	406	46.62	7.79	12/11	Nucleus
BraA03g021060.3.1C	AT2G39830	DAR2	BrDAR2.4	A03:10,048,359−10,051,673−	3315/1338	446	50.69	6.05	10/9	Endoplasmic reticulumn
BraA05g006160.3.5C	AT2G39830	DAR2	BrDAR2.5	A05:3,156,521−3,160,636+	4116/1320	440	50.36	8.36	9/8	Nucleus
BraA09g009490.3C	AT5G66640	DAR3	BrDAR3.1	A09:5,407,581−5,409,900−	2320/1551	517	59.32	6.1	9/8	Peroxisome
BraA07g017320.3C	AT5G66620	DAR6	BrDAR6.1	A07:14,995,298−14,998,531−	3234/2520	840	95.28	5.04	10/9	Nucleus
BraA09g009700.3.1C	AT5G66620	DAR6	BrDAR6.2	A09:5,411,650−5,414,348−	2699/1479	493	56.6	5.18	13/12	Nucleus
BraA09g009770.3.5C	AT5G66620	DAR6	BrDAR6.3	A09:5,411,699−5,414,567−	2869/462	154	17.11	6.43	3/2	Cytoplasm

## Data Availability

Not applicable.

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
