# Peer review of "Comprehensive In Silico Characterization and Expression Pro-Filing of DA1/DAR Family Genes in Brassica rapa"

_genes, 2022, doi:10.3390/genes13091577_

Round 1
Reviewer 1 Report
· Follow uniformity in representing units especially SI units throughout the MS
· Line 107: “16hight and 8h dark cycle” check this statement for correction
· Line 110 & 111: What is CK? PEG? Explain at its first cite
· Why the sample collection time differs only for temperature stress?
· Use small letter for representation of species eg. Brassica rapa not Brassica Rapa. Take care of this throughout the text
· In figure 5: In this caption, anaerobic stress was mentioned. But it was not explained in the materials and methods. Is it osmosis or anaerobic? Please check this. Similarly, check the caption of Figure 10. It is mentioned drought stress. If osmotic and drought stress are same use uniform terminology. Check these terms in the graph also
· Otherwise the results obtained are well explained and discussed
Author Response
Response to Reviewer 1 Comments
Point 1. Follow uniformity in representing units especially SI units throughout the MS
Response 1. Comment has been revised accordingly in MS (Table 1; Isoelectric Point (PI changed to SI unit pI)
Point 2. Line 107: “16hight and 8h dark cycle” check this statement for correction
Response 2. Comment has been revised accordingly in MS (Line 107)
Point 3. Line 110 & 111: What is CK? PEG? Explain at its first cite
Response 3. Comment has been revised accordingly in MS (Line 110, 111, 113, 114 & 116)
Point 4. Why the sample collection time differs only for temperature stress?
Response 4. Under temperature stress, Gene expressions are expected to be up or down-regulated during 24 hours, while under hormone and other stresses, gene expression differs quickly. That is why, we planned temperature stress for 24 hours with different suitable time intervals.
Point 5. Use small letter for representation of species eg. Brassica rapa not Brassica Rapa. Take care of this throughout the text
Response 5. Comment has been checked accordingly
Point 6. In figure 5: In this caption, anaerobic stress was mentioned. But it was not explained in the materials and methods. Is it osmosis or anaerobic? Please check this. Similarly, check the caption of Figure 10. It is mentioned drought stress. If osmotic and drought stress are same use uniform terminology. Check these terms in the graph also
Response 6. Comment has been revised accordingly in MS (Line 112)

Reviewer 2 Report
The manuscript of the article "Comprehensive in silico characterization and expression profiling of DA1/DAR family genes in Brassica rapa" by Umer Karamat, Rui Yang, Yuhong Ren, Yin Lu, Na Li and Jianjun Zhao is a small bioinformatic analysis of Brassica rapa sequences. Experimental work not presented. The study is empirical in nature. Perhaps it may be of some interest in planning experimental work. I think the authors should expand the introduction by formulating a clear statement of the problem. In addition, the discussion of the possibility of using data on the identification of certain hormone-related blocks should be expanded. What does this say, what exactly follows from this. Bioinformatics work without experimental work is rather difficult to correlate with real "wet" biology. How the authors see the application of their results in real practical science remains unclear.
Author Response
Response to Reviewer 2 Comments
Point 1. The manuscript of the article "Comprehensive in silico characterization and expression profiling of DA1/DAR family genes in Brassica rapa" by Umer Karamat, Rui Yang, Yuhong Ren, Yin Lu, Na Li and Jianjun Zhao is a small bioinformatic analysis of Brassica rapa sequences. Experimental work not presented. The study is empirical in nature. Perhaps it may be of some interest in planning experimental work.
Response 1: We performed expression analysis of BrDA1&DAR gene family under hormone treatment and abiotic stresses (Results; 3.8).
Point 2. I think the authors should expand the introduction by formulating a clear statement of the problem.
Response 2. Comment has been revised accordingly in MS (Line 50-52).
Point 3. In addition, the discussion of the possibility of using data on the identification of certain hormone-related blocks should be expanded. What does this say, what exactly follows from this. Bioinformatics work without experimental work is rather difficult to correlate with real "wet" biology. How the authors see the application of their results in real practical science remains unclear.
Response 3: Comment has been revised accordingly in MS (Line 340-347). According to your suggestions, the role of BrDA1&DAR gene family in the regulation of the exogenous application of hormones have been experimented and discussed. To correlate the bioinformatic work with the "wet" biology, we did QRT-PCR and discussed the role of ABA related cis-elements in the regulation of gene expression (Line 337-347). The identification of these genes for the studied stresses can be useful in developing the breeding program in Brassica rapa to create tolerance against studied stresses.
